# Convex Two-Layer Modeling with Latent Structure

**Vignesh Ganapathiraman**[†], **Xinhua Zhang**[†], **Yaoliang Yu**[∗], **Junfeng Wen**[♯]

[†]University of Illinois at Chicago, Chicago, IL, USA

[∗]University of Waterloo, Waterloo, ON, Canada, [♯]University of Alberta, Edmonton, AB, Canada

{vganap2, zhangx}@uic.edu, yaoliang.yu@uwaterloo.ca, junfengwen@gmail.com

## Abstract

Unsupervised learning of structured predictors has been a long standing pursuit in machine learning. Recently a conditional random field auto-encoder has been proposed in a two-layer setting, allowing latent structured representation to be automatically inferred. Aside from being nonconvex, it also requires the demanding inference of normalization. In this paper, we develop a convex relaxation of two-layer conditional model which captures latent structure and estimates model parameters, jointly and optimally. We further expand its applicability by resorting to a weaker form of inference—maximum a-posteriori. The flexibility of the model is demonstrated on two structures based on total unimodularity—graph matching and linear chain. Experimental results confirm the promise of the method.

## 1 Introduction

Over the past decade deep learning has achieved significant advances in many application areas [1]. By automating the acquisition of latent descriptive and predictive representation, they provide highly effective models to capture the relationships between observed variables. Recently more refined deep models have been proposed for structured output prediction, where several random variables for prediction are statistically correlated [2–4]. Improved performance has been achieved in applications such as image recognition and segmentation [5] and natural language parsing [6], amongst others.

So far, most deep models for structured output are designed for supervised learning where structured labels are available. Recently an extension has been made to the *unsupervised* learning. [7] proposed a conditional random field auto-encoder (CRF-AE)—a two-layer conditional model—where given the observed data $\mathbf{x}$, the latent structure $\mathbf{y}$ is first generated based on $p(\mathbf{y}|\mathbf{x})$, and then applied to reconstruct the observations using $p(\mathbf{x}|\mathbf{y})$. The motivation is to find the predictive and discriminative (rather than common but irrelevant) latent structure in the data. Along similar lines, several other discriminative unsupervised learning methods are also available [8–11].

Extending auto-encoders $X \to Y \to X$ to general two-layer models $X \to Y \to Z$ is not hard. [12, 13] addressed transliteration between two languages, where $Z$ is the observed binary label indicating if two words match, and higher accuracy can be achieved if we faithfully recover a letter-wise matching represented by the *unobserved* structure $Y$. In essence, their model optimizes $p(\mathbf{z}|\arg\max_{\mathbf{y}} p(\mathbf{y}|\mathbf{x}))$, uncovering the latent $\mathbf{y}$ via its mode under the first layer model. This is known as **bi-level** optimization because the $\arg\max$ of inner optimization is used. A soft variant adopts the mean of $\mathbf{y}$ [14]. In general, conditional models yield more accurate predictions than generative models $X - Y - Z$ (*e.g.* multi-wing harmoniums/RBMs), unless the latter is trained in a discriminative fashion [15].

In computation, all methods require certain forms of tractability in inference. CRF-AE leverages marginal inference on $p(\mathbf{y}|\mathbf{x})p(\mathbf{x}|\mathbf{y})$ (over $\mathbf{y}$) for EM. Contrastive divergence, instead, samples from $p(\mathbf{y}|\mathbf{x})$ [11]. For some structures like graph matching, neither of them is tractable [16, 17] (unless assuming first-order Markovian). In *single-layer* models, this challenge has been resolved by max-margin estimation, which relies only on the MAP of $p(\mathbf{y}|\mathbf{x})$ [18]. This oracle is much less demanding than sampling or normalization, as finding the most likely state can be much easier than summarizing over all possible $\mathbf{y}$. For example, MAP for graph matching can be solved by max-flow.

Unfortunately a direct extension of max-margin estimation to *two-layer* modeling meets with immediate obstacles, because here one has to solve $\max_{\mathbf{y}} p(\mathbf{y}|\mathbf{x})p(\mathbf{z}|\mathbf{y})$. In general, $p(\mathbf{z}|\mathbf{y})$ depends on $\mathbf{y}$ in a highly nonlinear form, making this augmented MAP inference intractable. This seems to leave the aforementioned bi-level optimization the only option that retains the sole dependency on MAP. However, solving this optimization poses substantial challenge when $\mathbf{y}$ is discrete, because the mode of $p(\mathbf{y}|\mathbf{x})$ is almost always invariant to small perturbations of model parameters (*i.e.* zero gradient).

In this paper we demonstrate that this optimization can be relaxed into a convex formulation while still preserving sufficient regularities to recover a non-trivial, nonlinear predictive model that supports structured latent representations. Recently a growing body of research has investigated globally trainable deep models. But they remain limited. [19] formulated convex conditional models using layer-wise kernels, connected through nonlinear losses. However these losses are data dependent, necessitating a transductive setting to retain the context. [20] used boosting but the underlying oracle is generally intractable. Specific global methods were also proposed for polynomial networks [21] and sum-product networks [22]. None of these methods accommodate structures in latent layers.

Our convex formulation is achieved by enforcing the first-order optimality conditions of inner level optimization via sublinear constraints. Using a semi-definite relaxation, we amount to the first two-layer model that allows latent structures to be inferred concurrently with model optimization while still admitting globally optimal solutions (§3). To the best of our knowledge, this is the first algorithm in machine learning that *directly* constructs a convex relaxation for a bi-level optimization based on the inner optimality conditions. Unlike [19], it results in a truly inductive model, and its flexibility is demonstrated with two example structures in the framework of total unimodularity (§4). The only inference required is MAP on $p(\mathbf{y}|\mathbf{x})$, and the overall scalability is further improved by a refined optimization algorithm (§5). Experimental results demonstrate its useful potentials in practice.

## 2  Preliminaries and Background

We consider a two-layer latent conditional model $X \to Y \to Z$, where $X$ is the input, $Z$ is the output, and $Y$ is a latent layer composed of $h$ random variables: $\{Y_i\}_{i=1}^h$. Instead of assuming no interdependency between $Y_i$ as in [19], our major goal here is to model the structure in the latent layer $Y$. Specifically, we assume a conditional model for the first layer based on an exponential family

$$p(\mathbf{y}|\mathbf{x}) = q_0(\mathbf{y})\exp(-\mathbf{y}'U\mathbf{x} - \Omega(U\mathbf{x})), \quad \text{where} \quad q_0(\mathbf{y}) = [\![\mathbf{y} \in \mathcal{Y}]\!]. \tag{1}$$

Here $U$ is the first layer weight matrix, and $\Omega$ is the log-partition function. $q_0(\mathbf{y})$ is the base measure, with $[\![x]\!] = 1$ if $x$ is true, and 0 otherwise. The correlation among $Y_i$ is instilled by the support set $\mathcal{Y}$, which plays a central role here. For example, when $\mathcal{Y}$ consists of all $h$-dimensional canonical vectors, $p(\mathbf{y}|\mathbf{x})$ recovers the logistic multiclass model. In general, to achieve a tradeoff between computational efficiency and representational flexibility, we make the following assumptions on $\mathcal{Y}$:

**Assumption 1** (PO-tractable)**.** *We assume $\mathcal{Y}$ is bounded, and admits an efficient polar operator. That is, for any vector $\mathbf{d} \in \mathbb{R}^h$, $\min_{\mathbf{y}\in\mathcal{Y}} \mathbf{d}'\mathbf{y}$ is efficiently solvable.*

Note the support set $\mathcal{Y}$ (hence the base measure $q_0$) is fixed and does *not* contain any more parameter. PO-tractability is available in a variety of applications, and we give two examples here.

**Graph matching.**  In a bipartite graph with two sets of vertices $\{a_i\}_{i=1}^n$ and $\{b_j\}_{j=1}^n$, each edge between $a_i$ and $b_j$ has a weight $T_{ij}$. The task is to find a one-to-one mapping (can be extended) between $\{a_i\}$ and $\{b_j\}$, such that the sum of weights on the edges is maximized. Denote the matching by $Y \in \{0,1\}^{n\times n}$ where $Y_{ij} = 1$ iff the edge $(a_i, b_j)$ is selected. So the optimal matching is the mode of $p(Y) \propto [\![Y \in \mathcal{Y}]\!]\exp(\text{tr}(Y'T))$ where the support is $\mathcal{Y} = \{Y \in \{0,1\}^{n\times n} : Y\mathbf{1} = Y'\mathbf{1} = \mathbf{1}\}$.

**Graphical models.**  For simplicity, consider a linear chain model $V_1 - V_2 - \cdots - V_p$. Here each $V_i$ can take one of $C$ possible values, which we encode using the $C$-dimensional canonical basis $\mathbf{v}_i$. Suppose there is a node potential $\mathbf{m}_i \in \mathbb{R}^C$ for each $V_i$, and each edge $(V_i, V_{i+1})$ has an edge potential $M_i \in \mathbb{R}^{C\times C}$. Then we could directly define a distribution on $\{V_i\}$. Unfortunately, it will involve quadratic terms such as $\mathbf{v}_i' M_i \mathbf{v}_{i+1}$, and so a different parameterization is in order. Let $Y_i \in \{0,1\}^{C\times C}$ encode the values of $(V_i, V_{i+1})$ via row and column indices of $Y_i$ respectively. Then the distribution on $\{V_i\}$ can be equivalently represented by a distribution on $\{Y_i\}$:

$$p(\{Y_i\}) \propto [\![\{Y_i\} \in \mathcal{Y}]\!]\exp\left(\sum_{i=1}^p \mathbf{m}_i'Y_i\mathbf{1} + \sum_{i=1}^{p-1}\text{tr}(M_i'Y_i)\right), \tag{2}$$

where $\mathcal{Y} = \left\{\{Y_i\} : Y_i \in \{0,1\}^{C\times C}\right\} \cap \mathcal{H}, \quad \text{with } \mathcal{H} := \left\{\{Y_i\} : \mathbf{1}'Y_i\mathbf{1} = 1, \, Y_i'\mathbf{1} = Y_{i+1}\mathbf{1}\right\}.$ (3)

The constraints in $\mathcal{H}$ encode the obvious consistency constraints between overlapping edges. This model ultimately falls into our framework in (1).

In both examples, the constraints in $\mathcal{Y}$ are totally unimodular (TUM), and therefore the polar operator can be computed by solving a linear programming (LP), with the $\{0, 1\}$ constraints relaxed to $[0, 1]$. In §4.1 and 4.2 we will generalize $\mathbf{y}'U\mathbf{x}$ to $\mathbf{y}'\mathbf{d}(U\mathbf{x})$, where $\mathbf{d}$ is an affine function of $U\mathbf{x}$ that allows for homogeneity in temporal models. For clarity, we first develop a general framework using $\mathbf{y}'U\mathbf{x}$.

**Output layer** As for the output layer, we assume a conditional model from an exponential family

$$p(\mathbf{z}|\mathbf{y}) = \exp(\mathbf{z}'R'\mathbf{y} - G(R'\mathbf{y}))q_1(\mathbf{z}) = \exp(-D_{G^*}(\mathbf{z}||\nabla G(R'\mathbf{y})) + G^*(\mathbf{z}))q_1(\mathbf{z}), \quad (4)$$

where $G$ is a smooth and strictly convex function, and $D_{G^*}$ is the Bregman divergence induced by the Fenchel dual $G^*$. Such a parameterization is justified by the equivalence between regular Bregman divergence and regular exponential family [23]. Thanks to the convexity of $G$, it is trivial to extend $p(\mathbf{z}|\mathbf{y})$ to $\mathbf{y} \in \text{conv}\mathcal{Y}$ (the convex hull of $\mathcal{Y}$), and $G(R'\mathbf{y})$ will still be convex over $\text{conv}\mathcal{Y}$ (fixing $R$).

Finally we highlight the assumptions we make and do not make. First we only assume PO-tractability of $\mathcal{Y}$, hence tractability in MAP inference of $p(\mathbf{y}|\mathbf{x})$. We do *not* assume it is tractable to compute the normalizer $\Omega$ or its gradient (marginal distributions). We also do not assume that unbiased samples of $\mathbf{y}$ can be drawn efficiently from $p(\mathbf{y}|\mathbf{x})$. In general, PO-tractability is a weaker assumption. For example, in graph matching MAP inference is tractable while marginalization is NP-hard [16] and sampling requires MCMC [24]. Finally, we do *not* assume tractability of any sort for $p(\mathbf{y}|\mathbf{x})p(\mathbf{z}|\mathbf{y})$ (in $\mathbf{y}$), and so it may be hard to solve $\min_{\mathbf{y}\in\mathcal{Y}}\{\mathbf{d}'\mathbf{y} + G(R'\mathbf{y}) - \mathbf{z}'R'\mathbf{y}\}$, as $G$ is generally not affine.

## 2.1 Training principles

At training time, we are provided with a set of feature-label pairs $(\mathbf{x}, \mathbf{z}) \sim \tilde{p}$, where $\tilde{p}$ is the empirical distribution. In the special case of auto-encoder, $\mathbf{z}$ is tied with $\mathbf{x}$. The "bootstrapping" style estimation [25] optimizes the joint likelihood with the latent $\mathbf{y}$ imputed in an optimistic fashion

$$\min_{U,R} \mathbb{E}_{(\mathbf{x},\mathbf{z})\sim\tilde{p}} \left[\min_{\mathbf{y}\in\mathcal{Y}} -\log p(\mathbf{y}|\mathbf{x})p(\mathbf{z}|\mathbf{y})\right] = \min_{U,R} \mathbb{E}_{(\mathbf{x},\mathbf{z})\sim\tilde{p}} \left[\min_{\mathbf{y}\in\mathcal{Y}} \mathbf{y}'U\mathbf{x} + \Omega(U\mathbf{x}) - \mathbf{z}'R'\mathbf{y} + G(R'\mathbf{y})\right].$$

This results in a hard EM estimation, and a soft version can be achieved by adding entropic regularizers on $\mathbf{y}$. Regularization can be imposed on $U$ and $R$ which we will make explicit later (*e.g.* bounding the $L_2$ norm). Since the log-partition function $\Omega$ in $p(\mathbf{y}|\mathbf{x})$ is hard to compute, the max-margin approach is introduced which replaces $\Omega(U\mathbf{x})$ by an upper bound $\max_{\hat{\mathbf{y}}\in\mathcal{Y}} -\hat{\mathbf{y}}'U\mathbf{x}$, leading to a surrogate loss

$$\min_{U,R} \mathbb{E}_{(\mathbf{x},\mathbf{z})\sim\tilde{p}} \left[\min_{\mathbf{y}\in\mathcal{Y}} \left\{-\mathbf{z}'R'\mathbf{y} + G(R'\mathbf{y}) + \mathbf{y}'U\mathbf{x} - \min_{\hat{\mathbf{y}}\in\mathcal{Y}} \hat{\mathbf{y}}'U\mathbf{x}\right\}\right]. \quad (5)$$

However, the key disadvantage of this method is the augmented inference on $\mathbf{y}$, because we have only assumed the tractability of $\min_{\mathbf{y}\in\mathcal{Y}} \mathbf{d}'\mathbf{y}$ for all $\mathbf{d}$, *not* $\min_{\mathbf{y}\in\mathcal{Y}}\{\mathbf{y}'\mathbf{d} + G(R'\mathbf{y}) - \mathbf{z}'R'\mathbf{y}\}$. In addition, this principle intrinsically determines the latent $\mathbf{y}$ as a function of *both* the input *and* the output, while at test time the output itself is unknown and is the subject of prediction. The common practice therefore requires a joint optimization over $\mathbf{y}$ and $\mathbf{z}$ at *test* time, which is costly in computation.

The goal of this paper is to design a convex formulation in which the latent $\mathbf{y}$ is completely determined by the input $\mathbf{x}$, and both the prediction and estimation rely only on the polar operator: $\arg\min_{\mathbf{y}\in\mathcal{Y}} \mathbf{y}'U\mathbf{x}$. As a consequence of this goal, it is natural to postulate that the $\mathbf{y}$ found this way renders an accurate prediction of $\mathbf{z}$, or a faithful recovery of $\mathbf{x}$ in auto-encoders. This idea, which has been employed by [e.g., 9, 26], leads to the following bi-level optimization problem

$$\max_{U,R} \mathbb{E}_{(\mathbf{x},\mathbf{z})\sim\tilde{p}} \left[\log p\left(\mathbf{z}\Big|\arg\max_{\mathbf{y}\in\mathcal{Y}} p(\mathbf{y}|\mathbf{x})\right)\right] \Leftrightarrow \max_{U,R} \mathbb{E}_{(\mathbf{x},\mathbf{z})\sim\tilde{p}} \left[\log p\left(\mathbf{z}\Big|\arg\min_{\mathbf{y}\in\mathcal{Y}} \mathbf{y}'U\mathbf{x}\right)\right] \quad (6)$$

$$\Leftrightarrow \min_{U,R} \mathbb{E}_{(\mathbf{x},\mathbf{z})\sim\tilde{p}} [-\mathbf{z}'R'\mathbf{y}_\mathbf{x}^* + G(R'\mathbf{y}_\mathbf{x}^*)], \quad \text{where } \mathbf{y}_\mathbf{x}^* = \arg\min_{\mathbf{y}\in\mathcal{Y}} \mathbf{y}'U\mathbf{x}. \quad (7)$$

Directly solving this optimization problem is challenging, because the optimal $\mathbf{y}_\mathbf{x}^*$ is almost surely invariant to small perturbations of $U$ (*e.g.* when $\mathcal{Y}$ is discrete). So a zero valued gradient is witnessed almost everywhere. Therefore a more carefully designed optimization algorithm is in demand.

## 3 A General Framework of Convexification

We propose addressing this bi-level optimization by convex relaxation, and it is built upon the first-order optimality conditions of the inner-level optimization. First notice that the set $\mathcal{Y}$ participates

in the problem (7) only via the polar operator at $U\mathbf{x}$: $\arg\min_{\mathbf{y}\in\mathcal{Y}} \mathbf{y}'U\mathbf{x}$. If $\mathcal{Y}$ is discrete, this problem is equivalent to optimizing over $S := \mathrm{conv}\,\mathcal{Y}$, because a linear function on a convex set is always optimized on its extreme points. Clearly, $S$ is convex, bounded, closed, and is PO-tractable. It is important to note that the origin is *not* necessarily contained in $S$. To remove the potential non-uniqueness of the minimizer in (7), we next add a small proximal term to the polar operator problem ($\sigma$ is a small positive number):

$$\min_{\mathbf{w}\in S} \mathbf{w}'U\mathbf{x} + \frac{\sigma}{2}\|\mathbf{w}\|^2. \tag{8}$$

This leads to a small change in the problem and makes sure that the minimizer is unique.[1] Adding strongly convex terms to the primal and dual objectives is a commonly used technique for accelerated optimization [27], and has been used in graphical model inference [e.g., 28]. We intentionally changed the symbol $\mathbf{y}$ into $\mathbf{w}$, because here the optimal $\mathbf{w}$ is not necessarily in $\mathcal{Y}$.

By the convexity of the problem (8) and noting that the gradient of the objective is $U\mathbf{x} + \sigma\mathbf{w}$, $\mathbf{w}$ is optimal *if and only if*

$$\mathbf{w} \in S, \quad \text{and} \quad (U\mathbf{x} + \sigma\mathbf{w})'(\hat{\boldsymbol{\theta}} - \mathbf{w}) \geq 0, \quad \forall \hat{\boldsymbol{\theta}} \in S. \tag{9}$$

These optimality conditions can be plugged into the bi-level optimization problem (7). Introducing "Lagrange multipliers" $(\gamma, \hat{\boldsymbol{\theta}})$ to enforce the latter condition via a mini-max formulation, we obtain

$$\min_{\|U\|\leq 1}\min_{\|R\|\leq 1}\mathbb{E}_{(\mathbf{x},\mathbf{z})\sim\tilde{p}}\Big[\min_{\mathbf{w}}\max_{\gamma\geq 0,\hat{\boldsymbol{\theta}}\in S}\max_{\mathbf{v}} -\mathbf{z}'R'\mathbf{w} + \mathbf{v}'R'\mathbf{w} - G^*(\mathbf{v}) \tag{10}$$

$$+\iota_S(\mathbf{w}) + \gamma(U\mathbf{x}+\sigma\mathbf{w})'(\mathbf{w}-\hat{\boldsymbol{\theta}})\Big], \tag{11}$$

where $\iota_S$ is the $\{0,\infty\}$-valued indicator function of the set $S$. Here we dualized $G$ via $G(R'\mathbf{w}) = \max_{\mathbf{v}} \mathbf{v}'R'\mathbf{w} - G^*(\mathbf{v})$, and made explicit the Frobenius norm constraints ($\|\cdot\|$) on $U$ and $R$.[2] Applying change of variable $\boldsymbol{\theta} = \gamma\hat{\boldsymbol{\theta}}$, the constraints $\gamma \geq 0$ and $\hat{\boldsymbol{\theta}} \in S$ (a convex set) become

$$(\boldsymbol{\theta},\gamma) \in \mathcal{N} := \mathrm{cone}\{(\hat{\boldsymbol{\theta}},1) : \hat{\boldsymbol{\theta}} \in S\},$$

where cone stands for the conic hull (convex). Similarly we can dualize $\iota_S(\mathbf{w}) = \max_{\boldsymbol{\pi}} \boldsymbol{\pi}'\mathbf{w} - \sigma_S(\boldsymbol{\pi})$, where $\sigma_S(\boldsymbol{\pi}) := \max_{\mathbf{w}\in S} \boldsymbol{\pi}'\mathbf{w}$ is the support function on $S$. Now swap $\min_{\mathbf{w}}$ with all the subsequent max (strong duality), we arrive at a form where $\mathbf{w}$ can be minimized out analytically

$$\min_{\|U\|\leq 1}\min_{\|R\|\leq 1}\mathbb{E}_{(\mathbf{x},\mathbf{z})\sim\tilde{p}}\Big[\max_{\boldsymbol{\pi}}\max_{(\boldsymbol{\theta},\gamma)\in\mathcal{N}}\max_{\mathbf{v}}\min_{\mathbf{w}} -\mathbf{z}'R'\mathbf{w} + \mathbf{v}'R'\mathbf{w} - G^*(\mathbf{v}) \tag{12}$$

$$+ \boldsymbol{\pi}'\mathbf{w} - \sigma_S(\boldsymbol{\pi}) + (U\mathbf{x}+\sigma\mathbf{w})'(\gamma\mathbf{w}-\boldsymbol{\theta})\Big] \tag{13}$$

$$= \min_{\|U\|\leq 1}\min_{\|R\|\leq 1}\mathbb{E}_{(\mathbf{x},\mathbf{z})\sim\tilde{p}}\Big[\max_{\boldsymbol{\pi}}\max_{(\boldsymbol{\theta},\gamma)\in\mathcal{N}}\max_{\mathbf{v}} -G^*(\mathbf{v}) - \sigma_S(\boldsymbol{\pi}) - \boldsymbol{\theta}'U\mathbf{x} \tag{14}$$

$$-\frac{1}{4\sigma\gamma}\|R(\mathbf{v}-\mathbf{z}) + \gamma U\mathbf{x} + \boldsymbol{\pi} - \sigma\boldsymbol{\theta}\|^2\Big]. \tag{15}$$

Given $(U,R)$, the optimal $(\mathbf{v},\boldsymbol{\pi},\boldsymbol{\theta},\gamma)$ can be efficiently solved through a concave maximization. However the overall objective is *not* convex in $(U,R)$ because the quadratic term in (15) is subtracted. Fortunately it turns out not hard to tackle this issue by using semi-definite programming (SDP) relaxation which linearizes the quadratic terms. In particular, let $I$ be the identity matrix, and define

$$M := M(U,R) := \begin{pmatrix} I \\ U' \\ R' \end{pmatrix}(I, U, R) = \begin{pmatrix} I & U & R \\ U' & U'U & U'R \\ R' & R'U & R'R \end{pmatrix} =: \begin{pmatrix} M_1 & M_u & M_r \\ M_u' & M_{u,u} & M_{r,u}' \\ M_r' & M_{r,u} & M_{r,r} \end{pmatrix}. \tag{16}$$

Then $\boldsymbol{\theta}'U\mathbf{x}$ can be replaced by $\boldsymbol{\theta}'M_u\mathbf{x}$ and the quadratic term in (15) can be expanded as

$$f(M,\boldsymbol{\pi},\boldsymbol{\theta},\gamma,\mathbf{v};\mathbf{x},\mathbf{z}) := \mathrm{tr}(M_{r,r}(\mathbf{v}-\mathbf{z})(\mathbf{v}-\mathbf{z})') + \gamma^2\,\mathrm{tr}(M_{u,u}\mathbf{x}\mathbf{x}') + 2\gamma\,\mathrm{tr}(M_{r,u}\mathbf{x}(\mathbf{v}-\mathbf{z})')$$

$$+ 2(\boldsymbol{\pi}-\sigma\boldsymbol{\theta})'(M_r(\mathbf{v}-\mathbf{z}) + \gamma M_u\mathbf{x}) + \|\boldsymbol{\pi}-\sigma\boldsymbol{\theta}\|^2. \tag{17}$$

Since given $(\boldsymbol{\pi},\boldsymbol{\theta},\gamma,\mathbf{v})$ the objective function becomes linear in $M$, so after maximizing out these variables the overall objective is convex in $M$. Although this change of variable turns the objective into convex, it indeed shifts the intractability into the feasible region of $M$:

$$\mathcal{M}_0 := \underbrace{\{M \succeq \mathbf{0} : M_1 = I, \mathrm{tr}(M_{u,u}) \leq 1, \mathrm{tr}(M_{r,r}) \leq 1\}}_{=:\mathcal{M}_1} \cap \{M : \mathrm{rank}(M) = h\}. \tag{18}$$

Here $M \succeq \mathbf{0}$ means $M$ is real symmetric and positive semi-definite. Due to the rank constraint, $\mathcal{M}_0$ is *not convex*. So a natural relaxation—the only relaxation we introduce besides the proximal term in (8)—is to drop this rank constraint and optimize with the resulting convex set $\mathcal{M}_1$. This leads to the final convex formulation:

$$\min_{M \in \mathcal{M}_1} \mathbb{E}_{(\mathbf{x},\mathbf{z}) \sim \tilde{p}} \left[ \max_{\boldsymbol{\pi}} \max_{(\boldsymbol{\theta},\gamma) \in \mathcal{N}} \max_{\mathbf{v}} -G^*(\mathbf{v}) - \sigma_S(\boldsymbol{\pi}) - \boldsymbol{\theta}' M_u \mathbf{x} - \frac{1}{4\sigma\gamma} f(M, \boldsymbol{\pi}, \boldsymbol{\theta}, \gamma, \mathbf{v}; \mathbf{x}, \mathbf{z}) \right]. \quad (19)$$

To summarize, we have achieved a convex model for two-layer conditional models in which the latent structured representation is determined by a polar operator. Instead of bypassing this bi-level optimization via the normal loss based approach [e.g., 19, 29], we addressed it directly by leveraging the optimality conditions of the inner optimization. A convex relaxation is then achieved via SDP.

## 3.1 Inducing low-rank solutions of relaxation

Although it is generally hard to provide a theoretical guarantee for nonlinear SDP relaxations, it is interesting to note that the constraint set $\mathcal{M}_1$ effectively encourages low-rank solutions (hence tighter relaxations). As a key technical result, we next show that all extreme points of $\mathcal{M}_1$ are rank $h$ (the number of hidden nodes) for all $h \geq 2$. Recall that in sparse coding, the atomic norm framework [30] induces low-complexity solutions by setting up the optimization over the convex hull of atoms, or penalize via its gauge function. Therefore the characterization of the extreme points of $\mathcal{M}_1$ might open up the possibility of analyzing our relaxation by leveraging the results in sparse coding.

**Lemma 1.** *Let $A_i$ be symmetric matrices. Consider the set of*

$$\mathcal{R} := \{X : X \succeq \mathbf{0}, \ \mathrm{tr}(A_i X) \lesseqgtr b_i, \ i = 1, \ldots, m\}, \quad (20)$$

*where $m$ is the number of linear (in)equality constraints. $\lesseqgtr$ means it can be any one of $\leq$, $=$, or $\geq$. Then the rank $r$ of all extreme points of $\mathcal{R}$ is upper bounded by*

$$r \leq \left\lfloor \tfrac{1}{2}(\sqrt{8m+1} - 1) \right\rfloor. \quad (21)$$

This result extends [31] by accommodating inequalities in (20), and its proof is given in Appendix A. Now we show that the feasible region $\mathcal{M}_1$ as defined by (18) has all extreme points with rank $h$.

**Theorem 1.** *If $h \geq 2$, then all extreme points of $\mathcal{M}_1$ have rank $h$, and $\mathcal{M}_1$ is the convex hull of $\mathcal{M}_0$.*

*Proof.* Let $M$ be an extreme point of $\mathcal{M}_1$. Noting that $M \succeq \mathbf{0}$ already encodes the symmetry of $M$, the linear constraints for $\mathcal{M}_1$ in (18) can be written as $\frac{1}{2}h(h+1)$ linear equality constraints and two linear inequality constraints. In total $m = \frac{1}{2}h(h+1) + 2$. Plug it into (21) in the above lemma

$$\mathrm{rank}(M) \leq \left\lfloor \tfrac{1}{2}(\sqrt{8m+1} - 1) \right\rfloor = \left\lfloor \tfrac{1}{2}(\sqrt{4h(h+1)+17} - 1) \right\rfloor = h + [\![h = 1]\!]. \quad (22)$$

Finally, the identity matrix in the top-left corner of $M$ forces $\mathrm{rank}(M) \geq h$. So $\mathrm{rank}(M) = h$ for all $h \geq 2$. It then follows that $\mathcal{M}_1 = \mathrm{conv}\mathcal{M}_0$. $\qquad \square$

## 4 Application in Machine Learning Problems

The framework developed above is generic. For example, when $\mathcal{Y}$ represents classification for $h$ classes by canonical vectors, $S = \mathrm{conv}\mathcal{Y}$ is the $h$ dimensional probability simplex (sum up to 1). Clearly $\sigma_S(\boldsymbol{\pi}) = \max_i \pi_i$, and $\mathcal{N} = \{(\mathbf{x}, t) \in \mathbb{R}_+^{h+1} : \mathbf{1}'\mathbf{x} = t\}$. In many applications, $\mathcal{Y}$ can be characterized as $\{\mathbf{y} \in \{0,1\}^h : A\mathbf{y} \leq \mathbf{c}\}$, where $A$ is TUM and all entries of $\mathbf{c}$ are in $\{-1, 1, 0\}$.[3] In this case, the convex hull $S$ has all extreme points being integral, and $S$ employs an explicit form:

$$\mathcal{Y} = \{\mathbf{y} \in \{0,1\}^h : A\mathbf{y} \leq \mathbf{c}\} \quad \Longrightarrow \quad S = \mathrm{conv}\mathcal{Y} = \{\mathbf{w} \in [0,1]^h : A\mathbf{w} \leq \mathbf{c}\}, \quad (23)$$

replacing all binary constraints $\{0,1\}$ by intervals $[0,1]$. Clearly TUM is a sufficient condition for PO-tractability, because $\min_{\mathbf{y} \in \mathcal{Y}} \mathbf{d}'\mathbf{y}$ is equivalent to $\min_{\mathbf{w} \in S} \mathbf{d}'\mathbf{w}$, an LP. Examples include the above graph matching and linear chain model. We will refer to $A\mathbf{w} \leq \mathbf{c}$ as the non-box constraint.

### 4.1 Graph matching

As the first concrete example, we consider convex relaxation for latent graph matching. One task in natural language is transliteration [12, 32]. Suppose we are given an English word $\mathbf{e}$ with $m$ letters, and a corresponding Hebrew word $\mathbf{h}$ with $n$ letters. The goal is to predict whether $\mathbf{e}$ and $\mathbf{h}$ are phonetically similar, a binary classification problem with $z \in \{-1, 1\}$. However it obviously helps to

find, as an intermediate step, the letter-wise matching between $\mathbf{e}$ and $\mathbf{h}$. The underlying assumption is that each letter corresponds to *at most* one letter in the word of the other language. So if we augment both $\mathbf{e}$ and $\mathbf{h}$ with a sink symbol * at the end (hence making their length $\hat{m} := m + 1$ and $\hat{n} := n + 1$ respectively), we would like to find a matching $\mathbf{y} \in \{0, 1\}^{\hat{m}\hat{n}}$ that minimizes the following cost

$$\min_{Y \in \mathcal{Y}} \sum_{i=1}^{\hat{m}} \sum_{j=1}^{\hat{n}} Y_{ij} \mathbf{u}' \boldsymbol{\phi}_{ij}, \text{ where } \mathcal{Y} = \{0, 1\}^{\hat{m} \times \hat{n}} \cap \underbrace{\{Y : Y_{i,:} \mathbf{1} = 1, \forall i \leq m, \ \mathbf{1}' Y_{:,j} = 1, \forall j \leq n\}}_{=: \mathcal{G}}. \quad (24)$$

Here $Y_{i,:}$ is the $i$-th row of $Y$. $\boldsymbol{\phi}_{ij} \in \mathbb{R}^p$ is a feature vector associated with the pair of $i$-th letter in $\mathbf{e}$ and $j$-th letter in $\mathbf{h}$, including the dummy *. Our notation omits its dependency on $\mathbf{e}$ and $\mathbf{h}$. $\mathbf{u}$ is a discriminative weight vector that will be learned from data. After finding the optimal $Y^*$, [12] uses the maximal objective value of (24) to make the final binary prediction: $-\operatorname{sign}(\sum_{ij} Y_{ij}^* \mathbf{u}' \boldsymbol{\phi}_{ij})$.

To pose the problem in our framework, we first notice that the non-box constraints $\mathcal{G}$ in (24) are TUM. Therefore, $S$ is simply $[0, 1]^{\hat{m} \times \hat{n}} \cap \mathcal{G}$. Given the decoded $\mathbf{w}$, the output labeling principle above essentially duplicates $\mathbf{u}$ as the output layer weight. A key advantage of our method is to allow the weights of the two layers to be *decoupled*. By using a weight vector $\mathbf{r} \in \mathbb{R}^p$, we define the output score as $\mathbf{r}' \Phi \mathbf{w}$, where $\Phi$ is a $p$-by-$\hat{m}\hat{n}$ matrix whose $(i, j)$-th column is $\boldsymbol{\phi}_{ij}$. So $\Phi$ depends on $\mathbf{e}$ and $\mathbf{h}$. Overall, our model follows by instantiating (12) as:

$$\min_{\|U\| \leq 1} \min_{\|R\| \leq 1} \mathbb{E}_{(\mathbf{e},\mathbf{h},z) \sim \tilde{p}} \Big[ \max_{\boldsymbol{\pi}} \max_{(\boldsymbol{\theta},\gamma) \in \mathcal{N}} \max_{v \in \mathbb{R}} \min_{\mathbf{w}} -z\mathbf{r}'\Phi\mathbf{w} + v\mathbf{r}'\Phi\mathbf{w} - G^*(v) + \boldsymbol{\pi}'\mathbf{w} \quad (25)$$

$$- \sigma_S(\boldsymbol{\pi}) + \sum\nolimits_{ij} (\mathbf{u}'\boldsymbol{\phi}_{ij} + \sigma w_{ij})(\gamma w_{ij} - \theta_{ij}) \Big]. \quad (26)$$

Once more we can minimize out $\mathbf{w}$, which gives rise to a quadratic $\|(v - z)\Phi'\mathbf{r} + \gamma\Phi'\mathbf{u} + \boldsymbol{\pi} - \sigma\boldsymbol{\theta}\|^2$. It is again amenable to SDP relaxation, where $(M_{u,u}, M_{r,u}, M_{r,r})$ correspond to $(\mathbf{u}\mathbf{u}', \mathbf{r}\mathbf{u}', \mathbf{r}\mathbf{r}')$ resp.

## 4.2 Homogeneous temporal models

A variety of structured output problems are formulated with graphical models. We highlight the gist of our technique by using a concrete example: unsupervised structured learning for inpainting. Suppose we are given images of handwritten words, each segmented into $p$ letters, and the latent representation is the corresponding letters. Since letters are correlated in their appearance in words, the recognition problem has long been addressed using linear chain conditional random fields. However imagine no ground truth letter label is available, and instead of predicting labels, we are given images in which a random small patch is occluded. So our goal will be inpainting the patches.

To cast the problem in our two-layer latent structure model, let each letter image in the word be denoted as a vector $\mathbf{x}_i \in \mathbb{R}^n$, and the reconstructed image be $\mathbf{z}_i \in \mathbb{R}^m$ ($m = n$ here). Let $Y_i \in \{0, 1\}^{h \times h}$ ($h = 26$) encode the labels of the letter pair at position $i$ and $i + 1$ (as rows and columns of $Y_i$ respectively). Let $U_v \in \mathbb{R}^{h \times n}$ be the letter-wise discriminative weights, and $U_e \in \mathbb{R}^{h \times h}$ be the pairwise weights. Then by (2), the MAP inference can be reformulated as (ref. definition of $\mathcal{H}$ in (3))

$$\min_{\{Y_i\} \in \mathcal{Y}} \sum_{i=1}^{p} \mathbf{1}' Y_i' U_v \mathbf{x}_i + \sum_{i=1}^{p-1} \operatorname{tr}(U_e' Y_i) \quad \text{where} \quad \mathcal{Y} = \big\{\{Y_i\} : Y_i \in \{0, 1\}^{C \times C}\big\} \cap \mathcal{H}. \quad (27)$$

Since the non-box constraints in $\mathcal{H}$ are TUM, the problem can be cast in our framework with $S = \operatorname{conv}\mathcal{Y} = \big\{\{Y_i\} : Y_i \in [0, 1]^{C \times C}\big\} \cap \mathcal{H}$. Finally to reconstruct the image for each letter, we assume that each letter $j$ has a basis vector $\mathbf{r}_j \in \mathbb{R}^m$. So given $W_i$, the output of reconstruction is $R'W_i\mathbf{1}$, where $R = (\mathbf{r}_1, \ldots, \mathbf{r}_h)'$. To summarize, our model can be instantiated from (12) as

$$\min_{\|U\| \leq 1} \min_{\|R\| \leq 1} \mathbb{E}_{(\mathbf{x},\mathbf{z}) \sim \tilde{p}} \Big[ \max_{\Pi} \max_{(\Theta,\gamma) \in \mathcal{N}} \max_{\mathbf{v}} \min_{W} \sum_{i=1}^{p} (\mathbf{v}_i - \mathbf{z}_i)' R' W_i \mathbf{1} - G^*(\mathbf{v}_i) \quad (28)$$

$$+ \operatorname{tr}(\Pi'W) - \sigma_S(\Pi) + \sum_{i=1}^{p} \operatorname{tr}((U_v \mathbf{x}_i \mathbf{1}' + [\![i \neq p]\!] U_e + \sigma W_i)'(\gamma W_i - \Theta_i)) \Big].$$

Here $\mathbf{z}_i$ is the inpainted images in the training set. If no training image is occluded, then just set $\mathbf{z}_i$ to $\mathbf{x}_i$. The constraints on $U$ and $R$ can be refined, e.g. bounding $\|U_v\|$, $\|U_e\|$, and $\|\mathbf{r}_j\|$ separately. As before, we can derive a quadratic term $\|R(\mathbf{v}_i - \mathbf{z}_i)\mathbf{1}' + \gamma U_v \mathbf{x}_i \mathbf{1}' + \gamma U_e + \Pi_i - \sigma\Theta_i\|^2$ by minimizing out $W_i$, which again leads to SDP relaxations. Even further, we may allow each letter to employ a *set* of principal components, whose combination yields the reconstruction (Appendix B).

Besides modeling flexibility, our method also accommodates problem-specific simplification. For example, the dimension of $\mathbf{w}$ is often much higher than the number of non-box constraints. Appendix C shows that for linear chain, the dimension of $\mathbf{w}$ can be reduced from $C^2$ to $C$ via partial Lagrangian.

# 5  Optimization

The key advantage of our convex relaxation (19) is that the inference depends on $\mathcal{S}$ (or equivalently $\mathcal{Y}$) *only through the polar operator*. Our overall optimization scheme is to perform projected SGD over the function of $M$. This requires: a) given $M$, compute its objective value and gradient; and b) project to $\mathcal{M}_1$. We next detail the solution to the former, relegating the latter to Appendix D.

Given $M$, we optimize over $(\boldsymbol{\pi}, \boldsymbol{\theta}, \gamma, \mathbf{v})$ by projected LBFGS [33]. The objective is easy to compute thanks to PO-tractability (for the $\sigma_S(\boldsymbol{\pi})$ term). The only nontrivial part is to project a point $(\boldsymbol{\theta}_0, \gamma_0)$ to $\mathcal{N}$, which is actually amenable to conditional gradient (CG). Formally it requires solving

$$\min_{\boldsymbol{\theta}, \gamma} \tfrac{1}{2}\|\boldsymbol{\theta} - \boldsymbol{\theta}_0\|^2 + \tfrac{1}{2}(\gamma - \gamma_0)^2, \quad s.t. \quad \boldsymbol{\theta} = \gamma\mathbf{s}, \ \gamma \in [0, C], \ \mathbf{s} \in S. \tag{29}$$

W.l.o.g., we manually introduced an upper bound[4] $C := \gamma_0 + \sqrt{\|\theta_0\|^2 + \gamma_0^2}$ on $\gamma$. At each iteration, CG queries the gradient $\mathbf{g}_{\boldsymbol{\theta}}$ in $\boldsymbol{\theta}$ and $g_\gamma$ in $\gamma$, and solves the polar operator problem on $\mathcal{N}$:

$$\min_{\boldsymbol{\theta} \in \gamma S, \gamma \in [0,C]} \boldsymbol{\theta}'\mathbf{g}_{\boldsymbol{\theta}} + \gamma g_\gamma = \min_{\mathbf{s} \in S, \gamma \in [0,C]} \gamma\mathbf{s}'\mathbf{g}_{\boldsymbol{\theta}} + \gamma g_\gamma = \min\{0, C\min_{\mathbf{s} \in S}(\mathbf{s}'\mathbf{g}_{\boldsymbol{\theta}} + g_\gamma)\}. \tag{30}$$

So it boils down to the polar operator on $S$, and is hence tractable. If the optimal value in (30) is nonnegative, then the current iterate is already optimal. Otherwise we add a basis $(\mathbf{s}^*, 1)$ to the ensemble and a totally corrective update can be performed by CG. More details are available in [34].

After finding the optimal $\hat{M}$, we recover the optimal $\mathbf{w}$ for each training example based on the optimal $\mathbf{w}$ in (12). Using it as the initial point, we locally optimize the two layer models $U$ and $R$ based on (14).

# 6  Experimental Results

To empirically evaluate our convex method (henceforth referred to as CVX), we compared it with the state-of-the-art methods on two prediction problems with latent structure.

**Transliteration**  The first experiment is based on the English-Hebrew corpus [35]. It consists of 250 positive transliteration pairs for training, and 300 pairs for testing. On average there are 6 characters per word in each of the languages. All these pairs are considered "positive examples", and for negative examples we followed [12] and randomly sampled $t_- \in \{50, 75, 100\}$ pairs from $250^2 - 250$ mismatched pairings (which are 20%, 30%, and 40% of 250, resp). We did not use many negative examples because, as per [12], our test performance measure will depend mainly on the highest few discriminative values, which are learned largely from the positive examples.

Given a pair of words $(\mathbf{e}, \mathbf{h})$, the feature representation $\phi_{ij}$ for the $i$-th letter in $\mathbf{e}$ and $j$-th letter in $\mathbf{h}$ is defined as the unigram feature: an $n$-dimensional vector with all 0's except a single one in the $(e_i, h_j)$-th coordinate. In this dataset, there are $n = 655$ possible letter pairs (* included). Since our primary objective is to determine whether the convex relaxation of a two-layer model with latent structure can outperform locally trained models, we adopted this simple but effective feature representation (rather than delving into heuristic feature engineering).

Our test evaluation measurement is the Mean Reciprocal Rank (MRR), which is the average of the reciprocal of the rank of the correct answer. In particular, for each English word $\mathbf{e}$, we calculated the discriminative score of respective methods when $\mathbf{e}$ is paired with each Hebrew word in the test set, and then found the rank of the correct word (1 for the highest). The reciprocal of the rank is averaged over all test pairs, giving the MRR. So a higher value is preferred, and 50% means on average the true Hebrew word is the runner-up. For our method, the discriminative score is simply $f := \mathbf{r}'\Phi\mathbf{w}$ (using the symbols in (25)), and that for [12] is $f := \max_{Y \in \mathcal{Y}} \mathbf{u}'\Phi\text{vec}(Y)$ (vectorization of $Y$).

We compared our method (with $\sigma = 0.1$) against the state-of-the-art approach in [12]. It is a special case of our model with the second-layer weight $\mathbf{r}$ tied with the first-layer weight $\mathbf{u}$. They trained it using a local optimization method, and we will refer to it as Local. Both methods employ an output loss function $\max\{0, yf\}^2$ with $y \in \{+1, -1\}$, and both contain only one parameter—the bound on $\|\mathbf{u}\|$ (and $\|\mathbf{r}\|$). We simply tuned it to optimize the performance of Local. The test MRR is shown in Figure 1, where the number of negative examples was varied in 50, 75, and 100. Local was trained with random initialization, and we repeated the random selection of the negative examples for 10 times, yielding 10 dots in each scatter plot. It is clear that CVX in general delivers significantly higher MRR than Local, with the dots lying above or close to the diagonal. Since this dataset is not big, the randomness of the negative set leads to notable variations in the performance (for both methods).

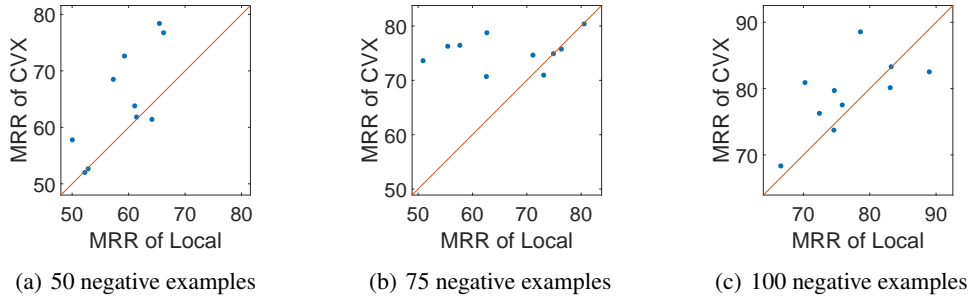

|  | (a) 50 negative examples | (b) 75 negative examples | (c) 100 negative examples |

Figure 1: MRR of Local versus CVX over 50, 75, and 100 negative examples.

| | SIZE OF OCCLUDED PATCH ($k \times k$) | | |
| --- | --- | --- | --- |
| | $k = 2$ | $k = 3$ | $k = 4$ |
| CRF-AE | $0.29 \pm 0.01$ | $0.80 \pm 0.01$ | $1.31 \pm 0.02$ |
| CVX | $0.27 \pm 0.01$ | $0.79 \pm 0.01$ | $1.28 \pm 0.02$ |

| | LENGTH OF SEQUENCE | | |
| --- | --- | --- | --- |
| | $p = 4$ | $p = 6$ | $p = 8$ |
| CRF-AE | $1.33 \pm 0.04$ | $1.30 \pm 0.02$ | $1.31 \pm 0.03$ |
| CVX | $1.29 \pm 0.04$ | $1.27 \pm 0.02$ | $1.28 \pm 0.03$ |

Table 1: Total inpainting error as a function of the size of occluded patch ($p = 8$).

Table 2: Total inpainting error as a function of the length of sequences ($k = 4$).

**Inpainting for occluded image**    Our second experiment used structured latent model to inpaint images. We generated 200 sequences of images for training, each with $p \in \{4, 6, 8\}$ digits. In order to introduce structure, each sequence can be either odd (*i.e.* all digits are either 1 or 3) or even (all digits are 2 or 4). So $C = 4$. Given the digit label, the corresponding image ($\mathbf{x} \in [0, 1]^{196}$) was sampled from the MNIST dataset, downsampled to 14-by-14. 200 test sequences were also generated.

In the test data, we randomly set a $k \times k$ patch of each image to 0 as occluded ($k \in \{2, 3, 4\}$), and the task is to inpaint it. This setting is entirely unsupervised, with no digit label available for training. It falls in the framework of $X \to Y \to Z$, where $X$ is the occluded input, $Y$ is the latent digit sequence, and $Z$ is the recovered image. In our convex method, we tied $U_v$ with $R$ and so we still have a 3-by-3 block matrix $M$, corresponding to $I$, $U_v$ and $U_e$. We set $\sigma$ to $10^{-1}$ and $G(\cdot) = \frac{1}{2} \|\cdot\|^2$ (Gaussian). $Y$ was predicted using the polar operator, based on which $Z$ was predicted with the Gaussian mean.

For comparison, we used CRF-AE, which was proposed very recently by [7]. Although it ties $X$ and $Z$, extension to our setting is trivial by computing the expected value of $Z$ given $X$. Here $P(Z|Y)$ is assumed a Gaussian whose mean is learned by maximizing $P(Z = \mathbf{x}|X = \mathbf{x})$, and we initialized all model parameters by unit Gaussian. For the ease of comparison, we introduced regularization by constraining model parameters to $L_2$ norm balls rather than penalizing the squared $L_2$ norm. For both methods, the radius bound was simply chosen as the maximum $L_2$ norm of the images, which produced consistently good results. We did not use higher $k$ because the images are sized 14-by-14.

The error of inpainting given by the two methods is shown in Table 1 where we varied the size of the occluded patch with $p$ fixed to 6, and in Table 2 where the length of the sequence $p$ was varied while $k$ was fixed to 4. Each number is the sum of squared error in the occluded patch, averaged over 5 random generations of training and test data (hence producing the mean and standard deviation). Here we can see that CVX gives lower error than CRF-AE. With no surprise, the error grows almost quadratically in $k$. When the length of sequence grows, the error of both CVX and CRF-AE fluctuates nonmonotonically. This is probably because with more images in each node, the total error is summed over more images, but the error per image decays thanks to the structure.

## 7    Conclusion

We have presented a new formulation of two-layer models with latent structure, while maintaining a jointly convex training objective. Its effectiveness is demonstrated by the superior empirical performance over local training, along with low-rank characterization of the extreme points of the feasible region. An interesting extension for future investigation is when the latent layer employs submodularity, with its base polytope mirroring the support set $S$.

## Footnotes

[1] If $p(\mathbf{y}|\mathbf{x}) \propto p_0(\mathbf{y})\exp(-\mathbf{y}'U\mathbf{x} - \frac{\sigma}{2}\|\mathbf{y}\|^2)$ (for any $\sigma > 0$), then there is no need to add this $\frac{\sigma}{2}\|\mathbf{w}\|^2$ term. In this case, all our subsequent developments apply directly. Therefore our approach applies to a broader setting where $L_2$ projection to $S$ is tractable, but here we focus on PO-tractability just for the clarity of presentation.

[2] To simplify the presentation, we bound the radius by 1 while in practice it is a hyperparameter to be tuned.

[3]For simplicity, we write equality constraints (handled separately in practice) using two inequality constraints.

[4]For $\gamma$ to be optimal, we require $(\gamma - \gamma_0)^2 \le \|\boldsymbol{\theta} - \boldsymbol{\theta}_0\|^2 + (\gamma - \gamma_0)^2 \le \|\mathbf{0} - \boldsymbol{\theta}_0\|^2 + (0 - \gamma_0)^2$, *i.e.*, $\gamma \le C$.

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
