[Supplementary Material]

## A Proof of Lemma 1

Let $X$ be an extreme point of $\mathcal{R}$ and $\text{rank}(X) = r$. We prove by contradiction, and suppose (21) does not hold. Then $\frac{1}{2}r(r+1) > m$. Write the eigen-decomposition of $X$ as $X = Q\Lambda Q'$, where $Q'Q = I$ and $\Lambda \in \mathbb{R}^{r \times r}$ is positive *definite* and diagonal. By (20), $\text{tr}(A_i X) = \text{tr}(A_i Q\Lambda Q') = \text{tr}(\Lambda Q' A_i Q) \lesseqqgtr b_i$.

Let $\text{S}^r$ be the set of symmetric $r$-by-$r$ matrices. It is a subspace with dimension $\frac{1}{2}r(r+1)$. Since $m < \frac{1}{2}r(r+1)$ and $Q'A_iQ \in \text{S}^r$, there must exist a nonzero $\Delta \in \text{S}^r$ such that $\text{tr}(\Delta Q' A_i Q) = 0$ for all $i$. In addition, as $\Lambda$ is positive *definite*, there exists $\epsilon > 0$ such that $\Lambda \pm \epsilon \Delta$ are both positive semi-definite. Now consider two matrices $X_+ = Q(\Lambda + \epsilon\Delta)Q'$ and $X_- = Q(\Lambda - \epsilon\Delta)Q'$. Clearly $X_+, X, X_-$ are distinct because $Q\Delta Q' = \mathbf{0}$ if, and only if, $\Delta = \mathbf{0}$. Furthermore $X_+$ and $X_-$ are both in $\mathcal{R}$ because i) $X_+ \succeq \mathbf{0}$ and $X_- \succeq \mathbf{0}$, and ii) $\text{tr}(A_i X_+) = \text{tr}(A_i X_-) = \text{tr}(A_i X)$. Now the fact that $X = (X_+ + X_-)/2$ contradicts with the assumption that $X$ is an extreme point of $\mathcal{R}$.

## B Chain model with sets of output bases

Finally to reconstruct the image for each letter, we assume that they are the convex combination of $p$ bases (or principal components) that are specific to each letter. Suppose for letter $j$, the bases are the columns of $R_j \in \mathbb{R}^{m \times p}$. Denote the combination weights for the $i$-th letter in a word as $Q_i \in \mathbb{R}_+^{p \times h}$, where the $j$-th column corresponds to the case where the letter is $j$. We postulate that $Q_i$ is related to $W_i$ in the sense that its $j$-th column is nonzero only if $W_i$ represents letter $j$: $Q_i'\mathbf{1} = W_i\mathbf{1}$. As a result, the expected reconstruction is $\sum_{j=1}^h R_j Q_i(:,j)$ where $Q_i(:,j)$ is the $j$-th column of $Q_i$. Enforcing this constraint with a Lagrange multiplier $\boldsymbol{\alpha}_i$, we finally obtain our objective

$$\min_{\|U\|\leq\lambda_1} \min_{\|R\|\leq\lambda_2} \mathbb{E}_{(\mathbf{x},\mathbf{z})\sim\tilde{p}} \Bigg[ \max_{\boldsymbol{\alpha}_i\geq\mathbf{0},\Pi} \max_{(\Theta,\gamma)\in\mathcal{C}} \max_{\mathbf{v}} \min_{W} \min_{Q_i\in[0,1]^{p\times h}} \tag{31}$$

$$\sum_i \left( (\mathbf{v}_i - \mathbf{z}_i)' \sum_j R_j Q_i(:,j) - G^*(\mathbf{v}_i) + \frac{\sigma'}{2}\|Q_i\|^2 \right) + \text{tr}(\Pi'W) - \sigma_S(\Pi) \tag{32}$$

$$+ \sum_i \text{tr}((U_v\mathbf{x}_i\mathbf{1}' + U_e + \sigma W_i)'(\gamma W_i - \Theta_i)) + \sum_i \boldsymbol{\alpha}_i'(Q_i'\mathbf{1} - W_i\mathbf{1})\Bigg]. \tag{33}$$

Here we added an extra small $L_2$ penalty on $Q_i$ and its weight $\sigma'$ is a small positive number. Then we proceed by

$$\min_{\|U\|\leq\lambda_1} \min_{\|R\|\leq\lambda_2} \mathbb{E}_{(\mathbf{x},\mathbf{z})\sim\tilde{p}} \Bigg[ \max_{\Pi} \max_{(\Theta,\gamma)\in\mathcal{C}} \max_{\mathbf{v}} -\sigma_S(\Pi) - \sum_i (G^*(\mathbf{v}_i) + \text{tr}(\Theta_i'(U_v\mathbf{x}_i\mathbf{1}' + U_e))) \tag{34}$$

$$- \frac{1}{4\sigma\gamma} \sum_i \|\gamma U_v\mathbf{x}_i\mathbf{1}' + \gamma U_e + \Pi_i - \sigma\Theta_i - \boldsymbol{\alpha}_i\mathbf{1}'\|^2 \tag{35}$$

$$+ \sum_i \min_{Q_i\in[0,1]^{p\times h}} \left\{ \frac{\sigma'}{2}\|Q_i\|^2 + (\mathbf{v}_i - \mathbf{z}_i)'\sum_j R_j Q_i(:,j) + \boldsymbol{\alpha}_i'Q_i'\mathbf{1} \right\}\Bigg]. \tag{36}$$

The last term is minimizing a quadratic form of $Q_i$ over $[0,1]$ constraints. This is exactly the same as we discussed in Section C. So following the same derivations there we get an SDP relaxation again.

An even more careful look reveals that the terms related to $U$ are only in (34) and (35), while the terms related to $R$ are only in (36). The absence of cross terms allows us to carry out SDP for $U$ and $R$ separately by considering matrices

$$\begin{pmatrix} I \\ U' \end{pmatrix}(I, U) = \begin{pmatrix} I & U \\ U' & U'U \end{pmatrix} \quad \text{and} \quad \begin{pmatrix} I \\ R' \end{pmatrix}(I, R) = \begin{pmatrix} I & R \\ R' & R'R \end{pmatrix}. \tag{37}$$

Note the technique of decoupling the hidden variables can also be applied to the more general framework in (12). The trade-off is delicate between the size of SDP size and the complexity of solving inner maximization given $M$. We conjecture that the SDP relaxation over $U$ and $R$ separately may lead to tighter approximation and higher sample efficiency. We leave the investigation for future work.

## C   Simplification via Partial Lagrangian Formulation

In many applications, the dimensionality of $\mathbf{w}$ is much higher than the number of non-box constraints. For example, in the homogeneous linear chain model, there are $O(C^2)$ variables (hence that number of $[0,1]$ box constraints), while the number of non-box constraints is $O(C)$. So a partial Lagrangian approach turns out more effective by retaining the box constraints in the optimization of $\mathbf{w}$ in (12), while the non-box constraints are enforced by Lagrange multipliers. This leads to the following objective that replaces the expression inside the expectation operator of (12):

$$\max_{\boldsymbol{\beta}\geq\mathbf{0}} \max_{(\boldsymbol{\theta},\gamma)\in\mathcal{N}} \max_{\mathbf{v}} \min_{\mathbf{w}\in[0,1]^h} -\mathbf{z}'R'\mathbf{w} + \mathbf{v}'R'\mathbf{w} - G^*(\mathbf{v}) + \boldsymbol{\beta}'(A\mathbf{w}-\mathbf{c}) + (U\mathbf{x}+\sigma\mathbf{w})'(\gamma\mathbf{w}-\boldsymbol{\theta})$$

$$= \max_{\boldsymbol{\beta}\geq\mathbf{0}} \max_{(\boldsymbol{\theta},\gamma)\in\mathcal{N}} \max_{\mathbf{v}} -G^*(\mathbf{v}) - \boldsymbol{\beta}'\mathbf{c} - \boldsymbol{\theta}'U\mathbf{x} - 2\gamma\sigma g\left(\frac{R(\mathbf{v}-\mathbf{z})+\gamma U\mathbf{x}+A'\boldsymbol{\beta}-\sigma\boldsymbol{\theta}}{-2\gamma\sigma}\right), \quad (38)$$

$$\text{where} \quad g(\mathbf{s}) = \frac{1}{2}\left\|\mathbf{s}\right\|^2 - \frac{1}{2}\left\|\left[\left|\mathbf{s}-\tfrac{1}{2}\mathbf{1}\right|-\tfrac{1}{2}\mathbf{1}\right]_+\right\|^2. \quad (39)$$

Here $[x]_+ = \max\{0,x\}$, and is taken elementwise along with absolute value. The expression of $g$ is derived from the fact that $\min_{w\in[0,1]} \frac{1}{2}w^2 - sw = \frac{1}{2}(d^2-2sd)$, where $d$ is the median of $\{s,0,1\}$.

Compared with (12) and (14), the $\boldsymbol{\beta}$ used here has much lower dimension than $\boldsymbol{\pi}$. In addition, $g(\mathbf{s})$ is convex and therefore given $(U,R)$, the optimal $(\boldsymbol{\beta},\boldsymbol{\theta},\gamma,\mathbf{v})$ can be solved efficiently. More interestingly, thanks to the expression of $g(\mathbf{s})$ in (39), which is a quadratic minus a convex function in $\mathbf{s}$, our SDP relaxation can be easily extended to this partial Lagrangian framework. In fact, just replace $\|\mathbf{s}\|^2$ by using an affine function of $M$, and then we obtain a convex objective in $(U,R)$.

Although the derivation of (38) is based on the generic form in (12), it is straightforward to apply the same technique to specialized formulations in §4.1 and 4.2.

## D   Projection to $\mathcal{M}_1$

The projection to $\mathcal{M}_1$ means solving for a given $\hat{M}$:

$$\min_{M} \tfrac{1}{2}||M-\hat{M}||^2, \quad s.t. \quad M\succeq\mathbf{0}, \; M_1=I, \; \text{tr}(M_{u,u})\leq 1, \; \text{tr}(M_{r,r})\leq 1. \quad (40)$$

To solve it efficiently, we resort to a partial Lagrangian approach with the last three constraints enforced by Lagrange multipliers $\Lambda\in\mathbb{R}^{h\times h}$, $\alpha\geq 0$, and $\beta\geq 0$, keeping the $M\succeq\mathbf{0}$ in closed form:

$$\max_{\alpha\geq 0,\beta\geq 0,\Lambda\in\mathbb{R}^{h\times h}} \min_{M\succeq\mathbf{0}} \tfrac{1}{2}||M-\hat{M}||^2 - \text{tr}(\Lambda'(M_1-I)) + \alpha(\text{tr}(M_{u,u})-1) + \beta(\text{tr}(M_{r,r})-1).$$

Given $(\Lambda,\alpha,\beta)$, the optimal $M$ has a closed form solution via eigenvalue thresholding.