[Reviews · NeurIPS 2016]

Reviewer 1

Summary

In this work the authors derive a convex approximation to a bi-level training objective for two-layer networks where the hidden middle layer has structure -- i.e., it must satisfy certain complex constraints. They show their derivation, discuss how it applies in two specific structured settings, and perform experiments on transliteration and image inpainting.

Qualitative Assessment

My main concern with this paper is that it makes many assumptions, adjustments, and approximations. As a result, it is difficult to be confident about what the resulting algorithm is actually doing. For instance, the obvious likelihood objective is replaced with a "bi-level" objective that does not seem to have any natural interpretation in an optimization context. Here, it is chosen simply because it makes the optimization simpler, which, while not inherently bad, deserves some discussion. Does this change the types of models learned? How? Similarly, an SDP relaxation is made but not characterized -- how should we value the convexity it affords versus the degraded fidelity? There are also some technical/clarity issues. For instance, the claim that optimizing over S is equivalent to optimizing over Y in line 132 is false, unless Y is constrained in some way -- for example, to be a subset of {0,1}^d. But as far as I can tell such a constraint is never stated. There are also a number of typos and grammar mistakes that make the paper more difficult to read. The experiments are interesting and show some positive results with respect to CRF-AE, but given that the goal of the paper is essentially to "convexify" CRF-AE, it would be nice to see the specifics of this, rather than just final performance numbers. For instance, does CVX achieve higher likelihood than CRF-AE? (It might, since it is convex and therefore find the global optimum. Or it might not, since it makes changes to the objective and relaxes the optimization to make it convex.) If it doesn't achieve higher likelihood, then how can we explain the positive results? Does it run faster? Also, it would be nice to see how these results compare to non-structured methods as well, to see how important structure is for these problems. Overall, I am left with a lot of questions. I think the method might be useful, but in my opinion it needs a more careful presentation and analysis to be convincing, as well as more detailed experiments. --- Post-feedback edit: The authors rightly noted in their feedback that my claim about line 132 is wrong, i.e., the claim in the paper is correct. My apologies. I've raised my technical quality score to a 3.

Confidence in this Review

2-Confident (read it all; understood it all reasonably well)


Reviewer 2

Summary

A convex training formulation for a two-layer model p(z|y)p(y|x), i.e., the model Z <- Y <- X with X denoting the input, Y the latent space, and Z the output, is proposed and investigated. Contrasting existing work, the authors only require tractable MAP inference on p(y|x), which is available for graph-matching type models or generally any integer linear program with a totally unimodular constraint set. A classical approach for two-layer models is the bi-level optimization based solution of p(z|\argmax_y p(y|x)). I.e., we first optimize the encoding model over y in an `inner' optimization, and infer the solution over the decoder in a second `outer' step. Training the parameters of those models is challenging due to the fact that small perturbations of the encoder model lead to the same prediction of the latent representation y, which is assumed to be discrete. Hence gradients for training the models are zero almost everywhere. The presented approach plugs the first-order optimality conditions of the `inner' optimization into the `outer' optimization, by using Lagrange multipliers and a saddle-point formulation. The resulting problem is non-convex in the training parameters, and a semi-definite programming (SDP) formulation is proposed, which yields a convex relaxation after dropping the rank constraint. Hence the authors state that learning depends on only the MAP operation over the latent space which is assumed to be efficient. The proposed approach is demonstrated on the task of transliteration and image inpainting. The method outperforms the shown baselines.

Qualitative Assessment

Review summary: A nice technique to circumvent the issues arising from bi-level optimization is suggested and applied to probabilistic modeling. However quite a few details are missing to assess the quality of the proposed solution, e.g., additional assumptions beyond the given ones might be required (see point 1 below), important aspects are not explained (see point 3 below), and the experimental evaluation is a little weak. See below for details. - Technical quality: derivations are sound but some additional assumptions should be required to ensure efficiency - Novelty: novel approach to two-layer modeling - Potential impact: hard to assess since details regarding efficiency of the proposed paper are not stated - Clarity: lacking explanation of some important aspects I'll adjust my score based on the author feedback. Review comments: 1. Let's assume z to be discrete. The decoder probability defined in Eq. (4) depends on the function G. To ensure proper normalization of the decoder distribution, G should correspond to the log-partition function. In the proposed formulation (e.g., Eq. (15)), maximization over its dual, i.e., the entropy restricted to the probability simplex, is required. Isn't tractability of this maximization task required/assumed then? In particular, the resulting program seems to be a quadratic term plus entropy terms and simplex constraints. Can the authors comment on how they solve this task efficiently? Sec. 5 does not seem to contain any details. 2. The following is not part of the proposed approach but the authors might nonetheless ensure correctness. In l.113, it is stated that the partition function `\Omega(Ux)' is replaced by an upper bound `\max -y'Ux'. The authors may want to check whether the maximization is indeed an `upper bound'. If I'm not wrong \Omega(Ux) = \log\sum_y\exp(-y'Ux) = -y^\ast'UX + \log\sum_y\exp(-y'Ux + y^\ast'Ux) (*) where y^\ast = \arg\max_y -y'Ux. However the log term in (*) can either be positive or negative, hence the maximization is not really a bound. Did I overlook something? 3. Why are the model parameters bounded? There were no constraints in the original formulation illustrated in Eq. (7). Hence it's counterintuitive to see constraints in Eq. (10) and the following ones. I think any reader would appreciate a more sophisticated argument beyond the stated one: `to simplify the presentation'. Could the authors provide a few more details regarding the reason for introducing those bound constraints? Is feasibility an issue? 4. Can the authors comment on the quality of the proposed baselines? The method is compared to an approach from 2010 on the transliteration task. It might be useful to provide more recent baselines as well. 5. The authors may also want to comment on the computational complexity of the proposed method. Usage of an SDP formulation seems computationally expensive and not necessarily scalable to larger problem sizes. 6. Moreoever I'm wondering whether the authors compared the proposed SDP approach to gradient descent on the objective given in Eq. (15). Minor comments: - The authors could mention that the ^\prime operator denotes the transpose.

Confidence in this Review

2-Confident (read it all; understood it all reasonably well)


Reviewer 3

Summary

This paper presents a two-layer model for learning problems with a latent structure. They first provide an objective and then discuss its convex relaxation by enforcing the first-order optimality condition via sublinear constraints. Experiments on two different tasks show the proposed approach is useful.

Qualitative Assessment

[Technical]; The paper provides nice analysis and makes the assumptions of their model clear. Empirical evaluations are also comprehensive. [novelty]: To my best knowledge the proposed approach in the paper is new. [Impact]: The proposed approach is general and may make a great impact on applications with latent structure. [Presentation]: Overall, the writing is nice. It provides a nice motivation in Section 2 and is not difficult to follow. Please see some comments below. Some comments in an arbitrary order: - Line 41-45: it is not clear what the authors mean in the introduction. Perhaps, provide an example? This statement is clearer after reading the whole paper. - (minor) I appreciate the authors summarize the literature of deep learning with structured prediction in the Introduction. However, it seems to me the proposed approach is not very relevant. Specifically, the model with only one latent layer and the relationship between x and y is encoded through y'Ux and it seems that the connection to the neural network model and auto-encoder are not strong. As a result, the introduction does not motivate the approach well (Section 2, in fact, provides a better introduction to the goal of this paper). - It seems to me the approach is relevant to [a], although [a] doesn't introduce structure into the latent layer. However, the joint likelihood approach and the convexification are relevant. - The authors may want to mention and relates their methods to latent structured prediction approaches including hidden CRF [b] and latent Structured SVM [c]. The key difference is in [b] and [c], the relations between x,y,z are directly encoded by features. [a] Convex Two-Layer Modeling [b] Hidden-state Conditional Random Fields [c] Learning Structural SVMs with Latent Variables *************************ADDITIONAL COMMENTS********************** Sorry for adding additional comments after the rebuttal period starts. However, it would be helpful if the authors can clarify some additional questions about the experiments: - [12] reported 95.4 in MRR, while the results reported in this paper are much lower. Then, I realize the experiment setting is slightly different. What is the performance of the proposed approach in the setting of [12]? - The authors said "[12] trained it using a local optimization method, and we will refer to it as Local". But [12] actually performed a global learning -- this is actually the main claim of [12]. I wonder if the authors implemented [12] right or if I misunderstand the authors.

Confidence in this Review

2-Confident (read it all; understood it all reasonably well)


Reviewer 4

Summary

The paper presents a technique for bi level modeling input -> latent -> output where the latent layer is generally a discrete (possibly combinatorial) structure. The set up is applicable to some problem like transliteration in NLP where the latent representation is the bipartite matching. The goal of the paper is to reduce the learning problem to a convex problem which can be solved efficiently provided any linear maximization subject to the latent space constraints (the polar operator) can be solved efficiently. The authors correctly point out that usually this problem is modeled as a bi-level optimization: first the latent layer is selected conditioned on the input and then the output is predicted conditioned on the latent layer. This paper massages the problem by relaxing the discrete latent variables into continuous variables. After using the properties of duality, the paper finally obtained a SDP formulation. With appropriate relaxations, the SDP can be convexified and solved efficiently.

Qualitative Assessment

The paper attacks a reasonably solid problem and the approach generally seems reasonable. I think the paper can benefit from better exposition of technical derivations. In particular, the derivations in page (4) can benefit greatly if the authors indicate what is the roadmap i.e. why are the derivations being made. Also I am not completely certain about some of the claims and would like clarifications for those: 1) Equation 19 contains maximization w.r.t \pi \in S. However the function being maximized is quadratic w.r.t. \pi -- how is that handled (since the assumption is only that linear functions over \s are tractable). 2) Also I think the authors erroneously claim that the rank relaxation is the only relaxation they introduce -- they also relax the latent variable 'y' from discrete space into continuous with the perturbed term, which is the key relaxation assumption. Please clarify.

Confidence in this Review

2-Confident (read it all; understood it all reasonably well)


Reviewer 5

Summary

This paper describes an approach to prediction with structured latent variable models (also applies to structured auto-encoders) that achieves convexity and tractability under one reasonable assumption (polar operator tractability) and one relaxation (having to do with rank in an SDP formulation).

Qualitative Assessment

The approach here is novel, technically deep and (apparently) sound, on a problem lots of smart people have thought about. The approach likely suffers scalability difficulties, due to the reduction to SDP, but overall this is a novel approach to a hard problem and I can imagine this generating substantial follow-on work. The paper is also remarkably clear, especially considering how technical it is in parts. One weakness of the writing is that it is not always clear what is a novel idea and what is an application of known ideas; I think part of this is the writing style. This could be easily clarified. There is one part that is a bit confusing, in this regard. The paper claims the only relaxation is the rank relaxation, but I don't think this is accurate. There is also the relaxation in Eq 5 where \Omega(Ux) is upper-bounded by max_y y'Ux, justified by a max-margin principle. This is fine, but if the goal is probabilistic models, it's not clear to me that this is entirely justified.

Confidence in this Review

2-Confident (read it all; understood it all reasonably well)


Reviewer 6

Summary

This paper proposes a new formulation of two-layer models with latent structure with maintaining a jointly convex training objective. The resulting formulation is nonconvex and tackled by using semi-definite programming (SDP) relaxation. The effectiveness is demonstrated by the superior empirical performance over local training.

Qualitative Assessment

The paper is technically sound. However, I do not find it reasonable to assume a conditional model based on an exponential family for the first layer and further replace the log-partition function, Omega, by an upper bound. In the end, the resulting formulation is further convexifed by SDP relaxations. I have a few unclarified issues: (a) By solving the SDP, which kind of solution did we get for the original problem including Omega? Is the solution feasible to the original problem or relaxed one? (b) Does Theorem 1 imply that the SDP relaxation gives the exact solution of Problem (15)? If not, finding a relaxed solution of the optimization problem that is obtained as an upper bound seems not reasonable.

Confidence in this Review

2-Confident (read it all; understood it all reasonably well)